# Linking functional and molecular mechanisms of host resilience to malaria infection

**Tsukushi Kamiya[1][†]\*, Nicole M Davis[2][†], Megan A Greischar[3], David Schneider[2], Nicole Mideo[1]**

[1]Department of Ecology and Evolutionary Biology, University of Toronto, Toronto, Canada; [2]Department of Microbiology and Immunology, Stanford University, Stanford, United States; [3]Department of Ecology and Evolutionary Biology, Cornell University, Ithaca, United States

**Abstract** It remains challenging to understand why some hosts suffer severe illnesses, while others are unscathed by the same infection. We fitted a mathematical model to longitudinal measurements of parasite and red blood cell density in murine hosts from diverse genetic backgrounds to identify aspects of within-host interactions that explain variation in host resilience and survival during acute malaria infection. Among eight mouse strains that collectively span 90% of the common genetic diversity of laboratory mice, we found that high host mortality was associated with either weak parasite clearance, or a strong, yet imprecise response that inadvertently removes uninfected cells in excess. Subsequent cross-sectional cytokine assays revealed that the two distinct functional mechanisms of poor survival were underpinned by low expression of either pro- or anti-inflammatory cytokines, respectively. By combining mathematical modelling and molecular immunology assays, our study uncovered proximate mechanisms of diverse infection outcomes across multiple host strains and biological scales.

**\*For correspondence:**
tsukushi.kamiya@mail.utoronto.ca

[†]These authors contributed equally to this work

**Competing interests:** The authors declare that no competing interests exist.

## Introduction

In human malaria, infection outcomes range widely from sub-clinical to fatal. While it is difficult to disentangle the factors contributing to this variation in resilience to malaria, host genetics is a major determinant (*Hernandez-Valladares et al., 2005*; *López et al., 2010*; *Hedrick, 2011*). Even in experimental rodent malaria infections, where environment, diet, and as many other factors as possible are highly controlled, different mouse strains infected with the same strain of *Plasmodium chabaudi* demonstrate remarkable variation in infection dynamics and malaria mortality. Among eight strains of inbred laboratory mice (129S1/SvImJ, A/J, C57BL/6, CAST/EiJ, NOD/ShiLtJ, NZO/HILtJ, PWK/PhJ, WSB/EiJ) that collectively span 90% of the mouse genetic diversity commonly used in laboratory experiments (*Roberts et al., 2007*), survival from malaria infection ranges from less than 5–100% (*Figure 1*). Underlying this survival variation are likely differences in functional properties of within-host ecology (i.e. parasite growth, parasite clearance, and replenishment of red blood cells [RBCs]), which are difficult to measure directly. However, longitudinal measurements of host health and parasite burden can inform processes of within-host ecology (i.e. RBCs and infected red blood cells (iRBCs), respectively, in malaria infections). For example, a 'disease curve' — longitudinal data of health and parasites plotted against each other in a phase plane — helps visualise the process of parasite growth, host sickness and recovery at the individual host level (*Figure 2*; *Schneider, 2011*). Furthermore, a mathematical model fitted to these data can predict particular functional mechanisms (e.g. parasite proliferation [*Mideo et al., 2011*], specific versus non-specific immunity [*Wale et al., 2019*] and dose-dependent host responses [*Haydon et al., 2003*; *Metcalf et al., 2011*;

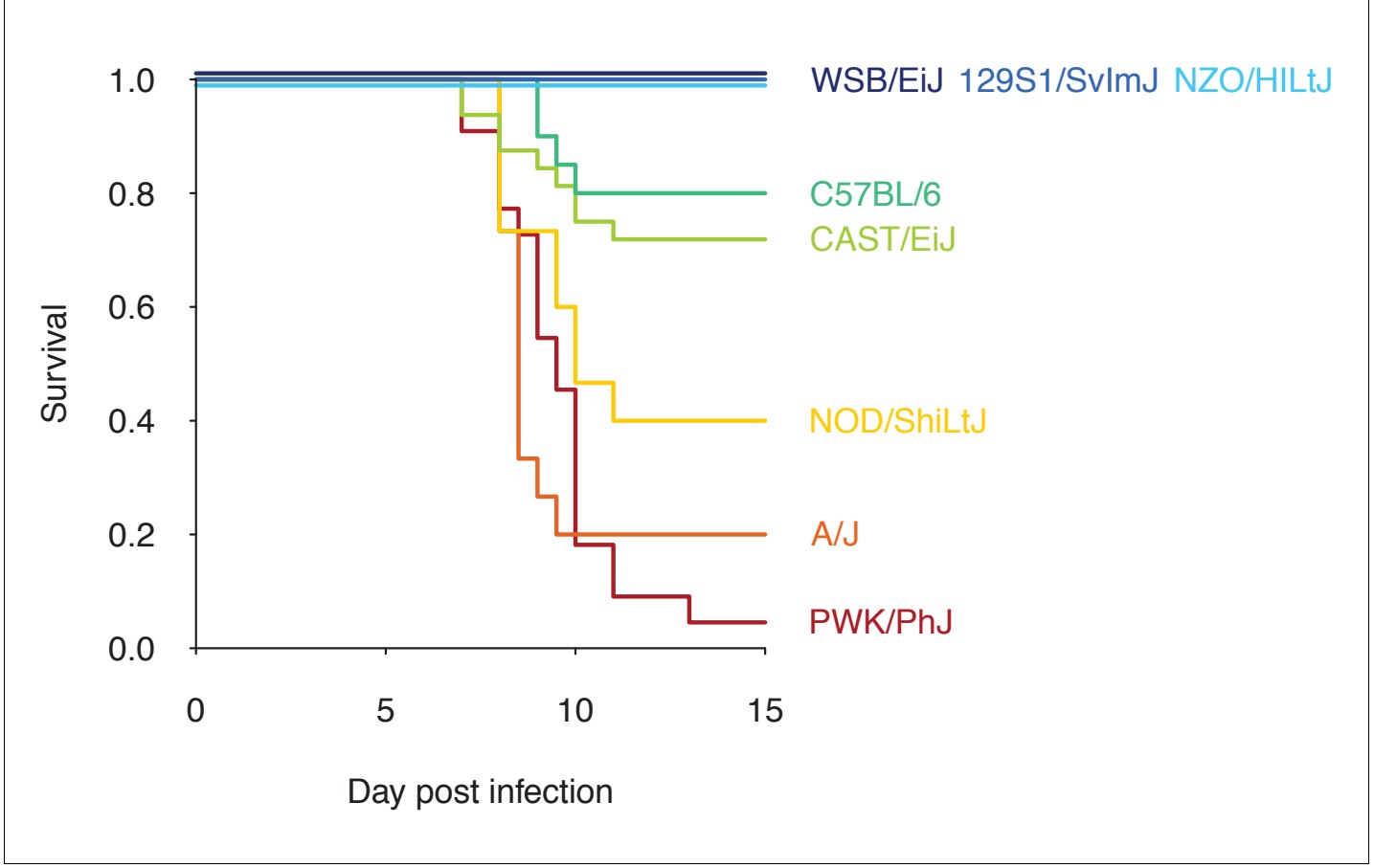

**Figure 1.** Proportion of mice surviving over the course of infections initiated with $10^5$ *P. chabaudi* AJ parasites. Eight mouse strains are shown (with total sample sizes given in parentheses): WSB/EiJ (30), 129S1/SvImJ (10), NZO/HILtJ (10), C57BL/6 (20), CAST/EiJ (32), NOD/ShiLtJ (15), A/J (15), and PWK/PhJ (22). The dataset is a compilation of two experiments (*Davis et al., 2021* and Gupta et al. unpublished). The lines for WSB/EiJ, 129S1/SvImJ and NZO/HILtJ are jittered as 100% of mice of each strain survived for 15 days.

*Kamiya et al., 2020*]) that lead to divergent infection dynamics (*Mideo et al., 2008a*; *Metcalf et al., 2012*). These model predictions can then be independently tested by follow-up experiments: for example, *Mideo et al., 2011* chemically manipulated RBC replenishment to interrogate the role of RBC age structure in parasite growth predicted by a previous model (*Mideo et al., 2008b*). Crucially, however, model predictions of host immune responses against malaria are rarely examined together with immunological data. Thus, it remains unclear whether and how modelled responses at the level of within-host ecology (e.g. rate of parasite clearance) are linked to observable quantities at the cellular and molecular levels (e.g. cytokines).

The immune system is a critical proximate mechanism of host genetic resilience to infection (*López et al., 2010*). Failure to mount a robust immune response can lead to unchecked parasite proliferation, while dysregulated responses may cause collateral damage, that is, immunopathology. While the benefit of immune protection often outweighs any costs associated with these responses (*Sorci et al., 2017*), severe outcomes of many infectious diseases are a consequence of immunopathology rather than direct damage caused by parasites (*Graham et al., 2005*). Thus, a 'healthy' immune response requires striking a delicate balance.

During the acute phase of blood-stage malaria infection, innate responses target and remove iRBCs as well as short-lived extracellular parasites known as merozoites (*Stevenson and Riley, 2004*). In addition, RBCs — regardless of infection status — are susceptible to clearance by immune effectors such as macrophages (*Jakeman et al., 1999*; *Chua et al., 2013*). While the targeted response removes more iRBCs, data-driven modelling studies highlight the importance of indiscriminate RBC clearance for lowering parasite burden (*Wale et al., 2019*; *Kamiya et al., 2020*;

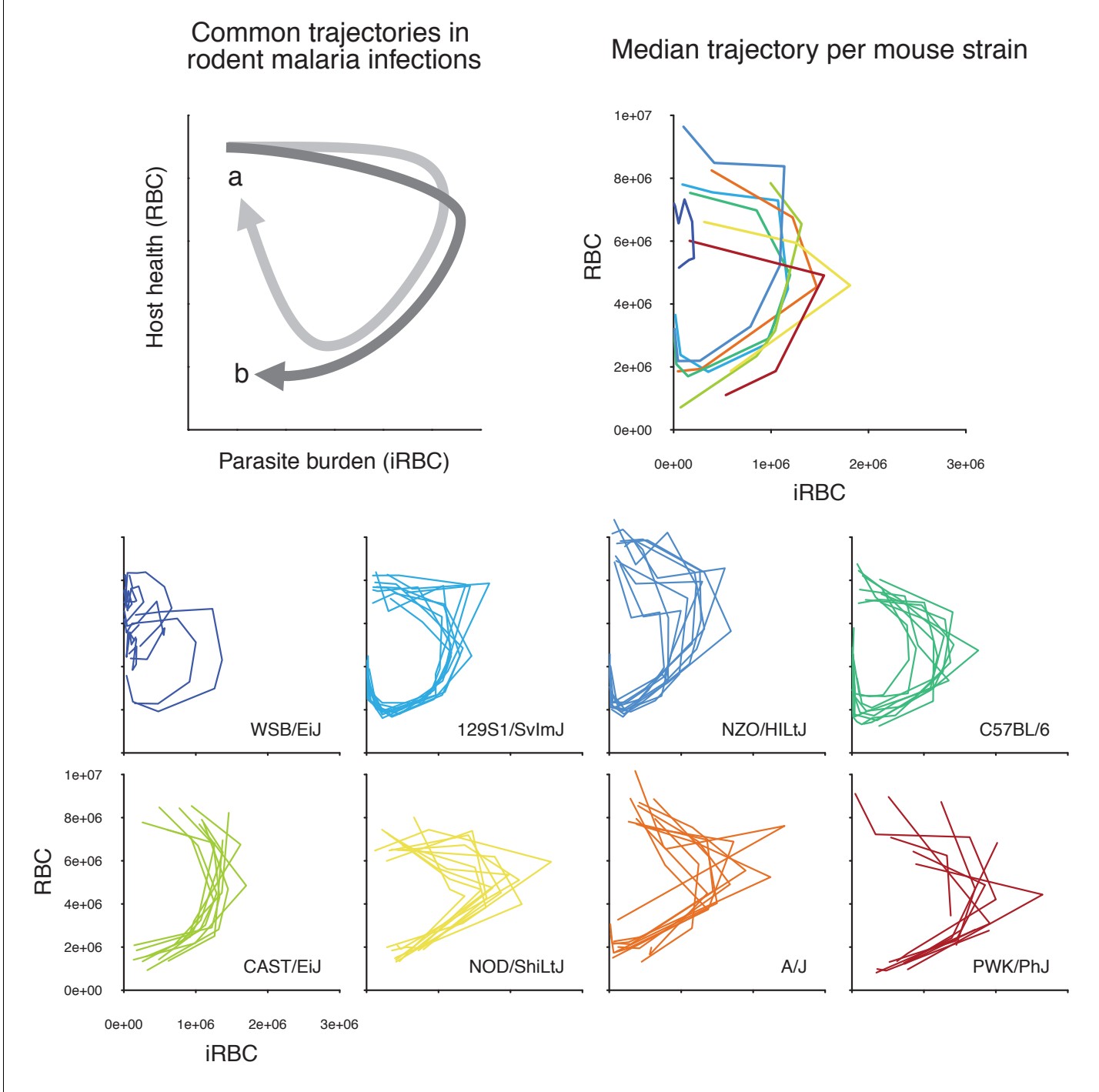

**Figure 2.** Longitudinal data of infection contain features of within-host ecology that influence infection outcomes. In a phase plot bound by parasite burden and host health (i.e. disease space, sensu [*Schneider, 2011*; *Torres et al., 2016*]), infection progresses clockwise from the top-left corner (i.e. many RBCs, few iRBCs). Top left panel illustrates common trajectories. Following a rapid parasite growth phase (rightward movement along the x-axis), host health deteriorates (downward movement along the y-axis) during acute malaria infection. In the meantime, the parasite density starts to decline due to resource limitation and/or upregulated immunity. If a host is resilient, the trajectory tends towards the starting healthy state as parasites further decline and RBCs are replenished (path a, light grey). In contrast, the damage to host health may be irreparable in non-resilient hosts (path b, dark grey) (*Schneider, 2011*). The small, coloured plots at the bottom show the empirically observed trajectories of the first wave of malaria infection in 80 mice across eight strains in disease space, with the densities of iRBCs and RBCs on the x- and y-axis, respectively. The top right panel shows the median trajectory of the eight strains. Generally speaking, highly resilient strains (WSB/EiJ, 129S1/SvImJ, NZO/HILtJ, C57BL/6) follow path a, and less resilient strains (CAST/EiJ, NOD/ShiLtJ, A/J, PWK/PhJ) follow path b.

*Metcalf et al., 2012*; *Miller et al., 2010*). Functionally, this host-driven destruction of RBCs can have both favourable and unfavourable consequences for host health. On the one hand, it has been proposed as a host adaptation in the presence of malaria parasites to clear the parasites directly (i.e. top-down effect) as well as to limit resources for the parasite (i.e. bottom-up effect) (*Wale et al., 2019*; *Haydon et al., 2003*; *Metcalf et al., 2012*; *Cromer et al., 2009*). On the other hand, an excessive loss of RBCs brings forth adverse health implications. In immune naive infants and children, severe malarial anaemia is the most common severe manifestation of disease, and its associated mortality rate can reach 30% (*Perkins et al., 2011*). A variety of processes cause malaria-related anaemia, including loss due to parasite exploitation, RBC clearance (e.g. phagocytosis of both infected and uninfected cells), suppression of RBC production, and defective RBC development (*Chua et al., 2013*). Among them, clearance is the most important process, accounting for between 75% and 90% of the total RBC deficit during malaria infections (*Jakeman et al., 1999*). In comparison, direct exploitation by malaria parasites has been estimated to account for less than 10% of the RBC deficit (*Jakeman et al., 1999*; *Price et al., 2001*; *Fonseca et al., 2016*).

At the molecular level, vertebrate host responses are regulated by immune signalling molecules, known as cytokines (*Lamb et al., 2006*). Acute malaria infection induces pro-inflammatory cytokines required for mounting a timely and robust response while anti-inflammatory cytokines inhibit excess immune reactions to safeguard against collateral damage (*Lamb et al., 2006*). For instance, tumour necrosis factor alpha ($\mathrm{TNF}-\alpha$) and interferon-gamma ($\mathrm{IFN}-\gamma$), are pro-inflammatory cytokines responsible for a myriad of inflammatory responses, including the production of nitric oxide and reactive oxygen species (*Bouharoun-Tayoun et al., 1995*; *Bogdan et al., 2000*), which are associated with rapid clearance of *P. falciparum*, the deadliest human malaria parasite (*Rockett et al., 1992*; *Kremsner et al., 1995*; *Mordmüller et al., 1997*; *Hernandez-Valladares et al., 2006*; *Franklin et al., 2007*). However, the same inflammatory responses can also be damaging to the organisms that produce them (*Clark et al., 1991*; *King and Lamb, 2015*). For example, TNF-α over-production — which can result from a deficit of anti-inflammatory cytokines like interleukin 10 (IL-10) and transforming growth factor-beta (TGF-β) — could lead to adverse effects including worsened anaemia, weight loss and survival in the mouse model (*Omer and Riley, 1998*; *Li et al., 2003*; *Long et al., 2006*; *Long et al., 2008*). Balanced expression of these cytokines is likely a mechanism that promotes resilience (survival) to malaria infection. However, mechanistic studies usually focus on just one or two inbred mouse strains with similar cytokine responses, limiting our ability to link molecular signatures with functional variation in host traits (e.g. indiscriminate versus targeted RBC clearance) that impacts the infection dynamics.

To uncover the functional mechanisms underlying malaria survival and variation thereof, we formulated a mathematical model of within-host malaria ecology that describes the asexual replicative cycle and qualitatively distinct components of host immunity (*Figure 3*). Rather than aiming for a mechanistically precise description of host immunity, we employed a simple mathematical model to track the net effects of host responses, that is, clearance rate of iRBCs and RBCs (*Kamiya et al., 2020*; *Figure 3a*). Using a hierarchical Bayesian approach, we fitted the model to longitudinal data of RBCs and iRBCs from eight mouse strains with varied resilience to *P. chabaudi*. We then examined cross-sectional cytokine data from the same eight strains to uncover the molecular underpinnings of our model predictions.

## Results and discussion

### Functional mechanisms underlying resilience to malaria

Our mathematical model of within-host malaria infection accurately described the time-course of RBCs and iRBCs during the acute phase of malaria infection in all mouse strains (Appendices 1 and 2). Several estimated model parameters varied with mouse strain (*Figure 4*). To characterise these multivariate, within-host ecological differences, we carried out principal component analysis (PCA) on the estimated parameter set, θ. We found several clusters that distinguished mouse strains revealing functional diversity of host resilience to malaria infection (*Figure 5*).

First, we identified C57BL/6 (80% survival; *Figure 1*) as the most 'functionally average' of the eight strains, indicated by the most central position in the PCA biplot (*Figure 5*) and near-zero estimates for strain-specificity, *s* (*Figure 4a*). Two fully resilient (100% survival) strains, 129S1/SvlmJ and

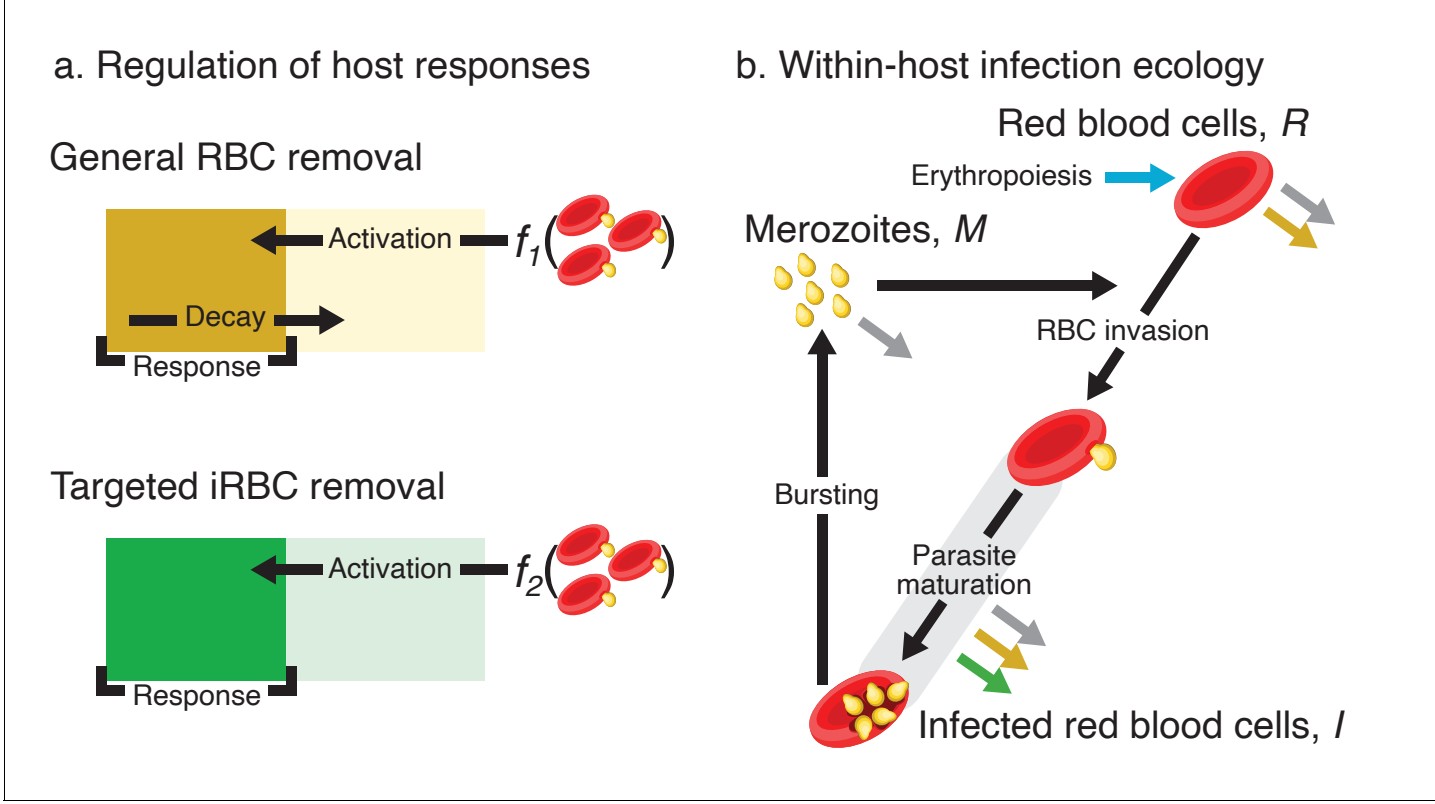

**Figure 3.** Model schematics. (**a**) A dynamical regulation model of host responses against blood-stage malaria. We condensed the complexity of the vertebrate acute innate response against malaria into two independent pathways responsible for general RBC clearance and targeted iRBC clearance (represented by the yellow and green block, respectively). Activation of each response occurs when the host detects the presence of pathogen-associated molecular patterns (PAMPs): $f_1$ and $f_2$ are linear functions of the iRBC density. For general RBC clearance, the activity resets daily. In contrast, the activity of the targeted responses against iRBCs accumulates over multiple days (see methods for further explanation). The output of each host response feeds back to influence the within-host infection dynamics (indicated by the coloured arrows in panel b). (**b**) Dynamics of RBCs and blood-stage malaria parasites within the host. Recruitment into and transitions among components of the asexual cycle are indicated with black arrows. Grey arrows indicate background mortality for different components. General clearance of RBCs and targeted clearance of iRBCs are marked with yellow and green arrows, respectively. Replenishment of RBCs (erythropoiesis) is indicated in blue.

NZO/HILtJ, were functionally similar to C57BL/6, but 129S1/SvImJ showed a slightly smaller parasite burst size, β, and higher background RBC clearance during infection, $\mu'_R$, while NZO/HILtJ showed slightly higher activation of both the indiscriminate response ($\psi_{N_1}$) and lower propensity to replenish RBCs, ρ (*Figure 4b*). These subtle functional differences contribute to suppressing parasite density and likely promote better resistance and survival outcomes, at least against this particular parasite genotype (*Figure 1*).

In addition to having the highest propensity to activate targeted clearance of iRBCs (highest $\psi_{N_2}$), WSB/EiJ, another fully resilient strain, demonstrated the smallest burst size, β (*Figure 4*). While we found a general negative association between β and host survival (*Figure 4*) — with a notable exception of PWK/PhJ mice — little is documented on the host's contribution to variation in iRBC burst size. One host factor that could affect parasite burst size is intrinsic differences in RBC properties among mouse strains. For example, the flow cytometry marker, TER119, a standard marker for mouse erythroid cells, works poorly with WSB/EiJ, hinting at a possible difference in RBC surface proteins (Davis, personal observation). WSB/EiJ also showed the highest capacity to increase background RBC mortality during infection, $\mu'_R$, which contributes to limiting parasite growth through lowered resource availability. Overall, WSB/EiJ excelled in every facet of resilience against *P. chabaudi* AJ and maintained comparatively much lower iRBC densities (*Figure 2*). Infection resilience in this mouse strain may generalise to other malaria parasites, including *P. berghei* (*Bopp et al., 2010*), yet interestingly, these mice are highly vulnerable to *Salmonella* infections (*Zhang et al., 2019*).

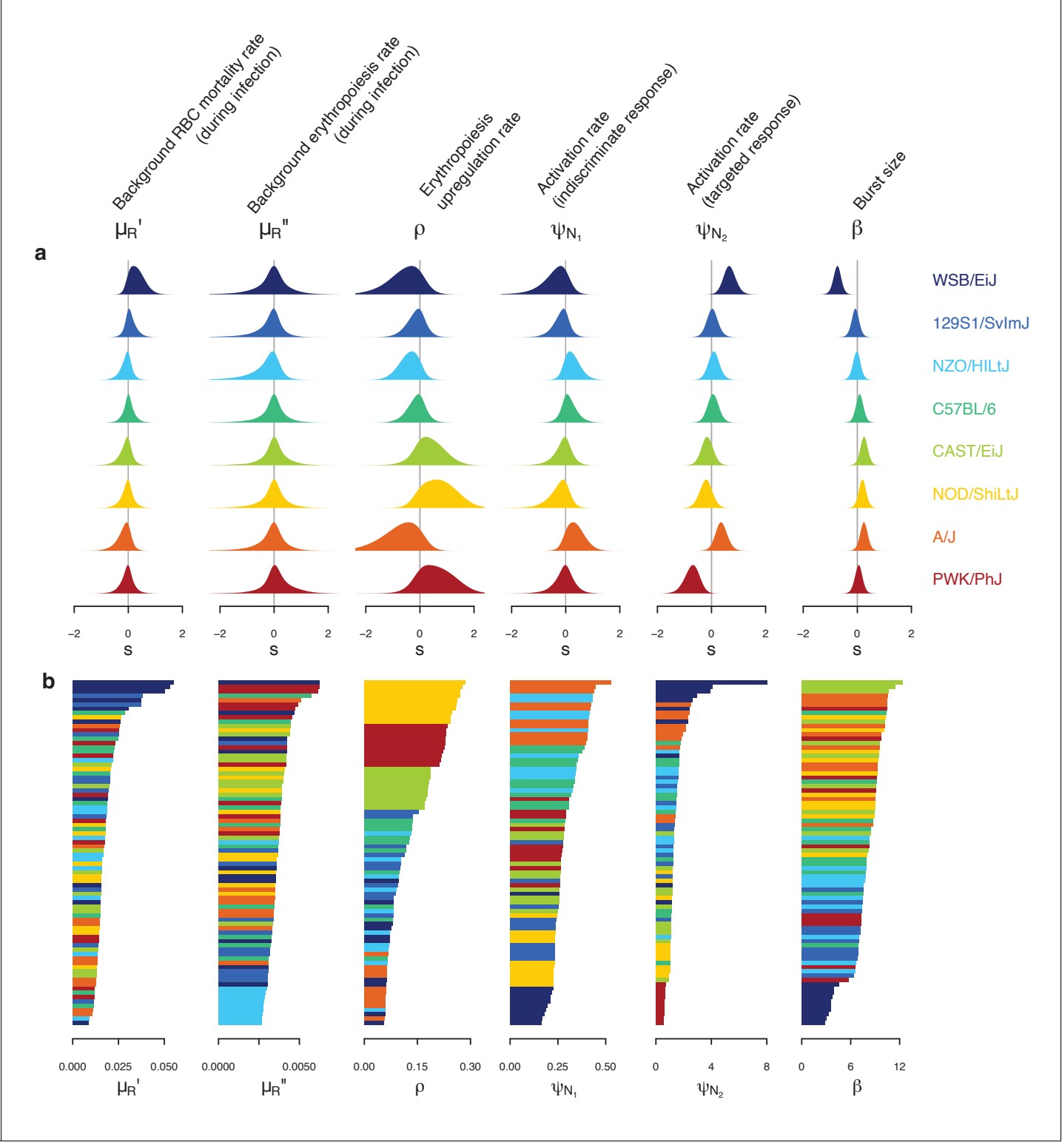

**Figure 4.** Differences in within-host ecological parameters reveal functional diversity linked to resilience to malaria infection. (a) Strain-specific variation, $s$, in each parameter of the set $\theta \ni (\mu_R', \mu_R'', \rho, \psi_{N_1}, \psi_{N_2}, \beta)$. The eight strains are ordered according to overall survival percentage from the top (see *Figure 1*). The average parameter value across the eight strains is indicated by $s = 0$. (b) Ordered parameter stacks show functional similarities and differences between individual mice of different strains (indicated by colours). Each slice of a stack represents the median estimate for an individual mouse.

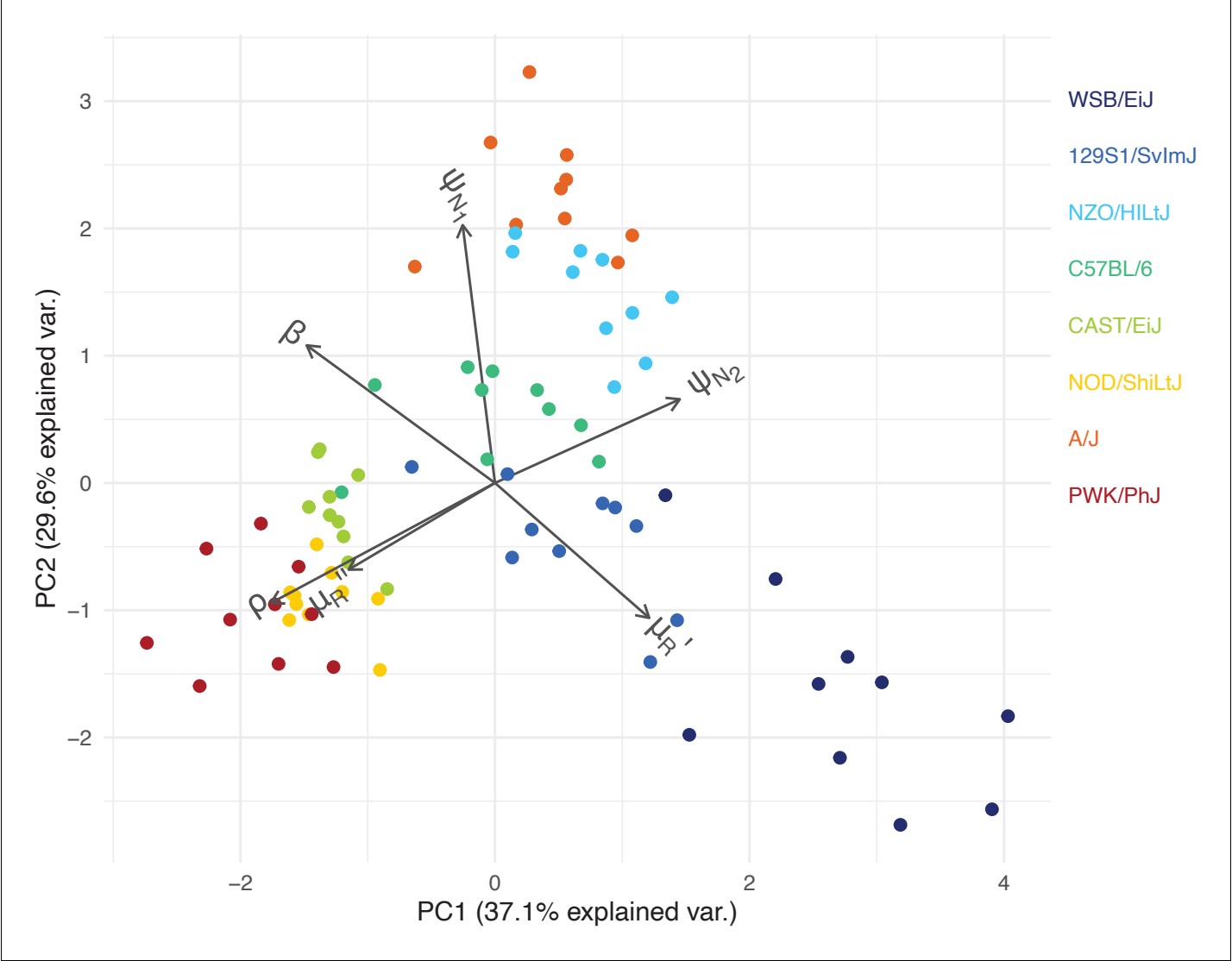

**Figure 5.** Within-host ecological parameters differentiate mouse strains with varying degrees of resilience to malaria infection. The PCA biplot displays the relationship between individual mice in the first two principal components, which collectively account for 66.7% of the total variance in parameters describing within-host malaria ecology, $\theta \ni (\mu_R', \mu_R'', \rho, \psi_{N_1}, \psi_{N_2}, \beta)$. The direction and length of the grey arrows indicate the contribution of each parameter to the principal components. Parameter descriptions are found in *Table 1*.

Three less resilient strains (CAST/EiJ, NOD/ShiLtJ, and PWK/PhJ) clustered together in the PCA biplot (*Figure 5*), indicating their functional similarity. We found a combination of top-down (immune-mediated) and bottom-up (resource-mediated) factors that distinguish these strains from the rest. Specifically, they showed low propensity to trigger a targeted immune response (low $\psi_{N_2}$, which hinders effective parasite clearance) and high erythropoiesis during infection (high $\rho$ and $\mu_R''$, which may inadvertently fuel parasite growth; *Figures 4* and *5*). Distinguishing moderately poor surviving strains (CAST/EiJ and NOD/ShiLtJ; 72% and 40% survival, respectively) from an extremely fragile strain (PWK/PhJ; <5% survival) is likely the markedly lower immune activation, $\psi_{N_2}$, in the latter, since other traits were similar among the three strains.

Finally, the A/J mouse strain showed the strongest activation of indiscriminate RBC clearance, $\psi_{N_1}$ and the second-highest activation of the targeted response, $\psi_{N_2}$ (*Figure 4b*). Given the poor resilience of these mice (20% survival; *Figure 1*), general RBC clearance is likely harmful, at least in this strain of mice. While the host potentially stands to benefit from destroying RBCs by removing some iRBCs and taking the resource away from malaria parasites, this 'scorched-earth tactic' (*Wale et al.,*

**Table 1.** Description of model parameters and their fixed values, or prior distributions used in Bayesian statistical inference. Where parameters were estimated (indicated by * on the description), we assigned generic priors (for immune parameters, $\psi_{N_1}$ and $\psi_{N_2}$, and hyperpriors $\sigma_s$ and $\sigma_u$) and weakly informative priors centred around specific estimates from previous studies for the rest.

| Symbol | Description | Fixed value or prior | Source |
|---|---|---|---|
| Host responses | | | |
| $\rho$ | Proportion of deviation from $R_c$ restored per day * | $0.25 \times \exp(\mathcal{N}(0,1))$ | *Miller et al., 2010* |
| $\psi_{N_1}$ | Activation strength of indiscriminate RBC clearance * | $\exp(\mathcal{N}(0,1))$ | |
| $\psi_{N_2}$ | Activation strength of targeted iRBC clearance * | $\exp(\mathcal{N}(0,1))$ | |
| Within-host infection dynamics | | | |
| $R_c$ | RBC density at homeostatic equilibrium | $\mathrm{RBC}_{(t=0)}$ | data |
| $I_{\max}$ | Maximum iRBC density observed | $2.65 \times 10^6$ per microliter | data |
| $\mu_R$ | Daily background RBC mortality rate | 0.025 | *Miller et al., 2010* |
| $\mu_R'$ | Daily background RBC mortality rate (during infection) * | $0.025 \times \exp(\mathcal{N}(0,1))$ | *Miller et al., 2010* |
| $\mu_R''$ | Density-independent RBC replenishment rate (during infection) * | $0.025 \times \exp(\mathcal{N}(0,1))$ | *Miller et al., 2010* |
| $\beta$ | Parasite burst size * | $7 \times \exp(\mathcal{N}(0,1))$ | *Miller et al., 2010* |
| $p$ | Merozoite invasion rate | $1.5 \times 10^{-5}$ per day | *Mideo et al., 2011* |
| $\mu_M$ | Merozoite mortality rate | 48 per day | *McAlister, 1977* |
| Hyperpriors | | | |
| $\sigma_s$ | Standard deviations for strain-level variation | $\exp(\mathcal{N}(0,1))$ | |
| $\sigma_u$ | Standard deviations for individual-level variation | $\exp(\mathcal{N}(0,1))$ | |
| Measurement errors | | | |
| $\sigma_{\mathrm{RBC}}$ | Standard deviation for total RBC density * | $5 \times 10^5 \times \exp(\mathcal{N}(0,1))$ | *Miller et al., 2010* |
| $\sigma_{\mathrm{iRBC}}$ | Standard deviation for $log_{10}$ iRBC count * | $0.2 \times \exp(\mathcal{N}(0,1))$ | *Mideo et al., 2008b* |

*2019*) could remove healthy RBCs in excess and trigger severe anaemia that causes host mortality. The significant role of indiscriminate RBC clearance on severe anaemia is empirically supported by a study that demonstrated a high turnover of transfused RBCs in BALB/c mice infected with *P. berghei* (*Evans et al., 2006*). The clearance was likely immune-mediated as severe anaemia was alleviated by depletion of immune cells (*Evans et al., 2006*). Another study observed lower young RBC (i.e., reticulocyte) counts in A/J mice and postulated that these mice are defective in the production of new RBCs during malaria infection (*Chang et al., 2004*). This is consistent with our model prediction that, among poorly resilient strains, A/J mice showed the lowest propensity to upregulate erythropoiesis (lower $\rho$ values, relative to CAST/EiJ, NOD/ShiLtJ, and PWK/PhJ; *Figure 4*). Together with our finding that A/J mice mount a stronger response than other poorly resilient strains (*Figure 4*), the lower $\rho$ values in A/J mice may reflect the inhibitory effect of inflammation on the steady-state erythropoiesis that takes place in the bone marrow (*Morceau et al., 2009*). As the interactions between inflammation and erythropoiesis are complex (e.g., inflammatory responses also induce stress erythropoiesis in the spleen [*Paulson et al., 2020*] and erythropoietin [EPO, cytokine primarily responsible for RBC production] inhibits inflammation [*Nairz et al., 2012*]), further investigation is needed to better understand the net impact of malaria-induced immune responses on RBC production.

The potentially negative impact of RBC clearance highlights vital implications for clinical interventions against malaria. First, blood transfusion or EPO injection to replenish depleted RBCs in severe anaemia may be most effective against patients that are particularly prone to indiscriminate RBC clearance. In fact, timely EPO injection alleviates anaemia and improves survival in A/J mice (*Chang et al., 2004*). However, the same treatment may be less effective against other poorly resilient patients (similar to CAST/EiJ, NOD/ShiLtJ, and PWK/PhJ) whose mortality is attributed to insufficient immune responses (*Figures 4* and *5*). Second, the potentially pathological consequence of indiscriminate RBC clearance should be considered during the development of a blood-stage malaria vaccine. Alarmingly, the possibility of immunopathology has so far been largely overlooked

in the vaccine development process (*Stanisic and Good, 2016*). It is pertinent to ensure that vaccine-triggered immunity that helps clear malaria parasites also avoids immunopathology, including severe anaemia. In summary, our model predicted that poor survival was associated with hosts with the weakest activation of the targeted response (in CAST/EiJ, NOD/ShiLtJ and PWK/PhJ), or the strongest activation of the indiscriminate response (in A/J). Thus, the host's ability to mount a precise response to clear parasites is likely a major determinant of host survival.

## Cytokine assays uncover molecular variation in immune responses

As our mathematical model identified the strength and precision of host immunity as the key functional motif of malaria resilience, we carried out a separate cross-sectional assay (with destructive sampling) and characterised the expression patterns of pro- and anti-inflammatory cytokines (i.e. immune signalling molecules), which play a pivotal role in regulating immune responses (*Figure 6*). Notably, pro-inflammatory cytokines such as TNF-α and IFN-γ impact malaria parasite clearance, while anti-inflammatory cytokines like IL-10 and TGF-β are crucial for limiting inflammation and immunopathology (*Artavanis-Tsakonas et al., 2003*).

In general, resilient strains (WSB/EiJ, 129S1/SvImJ, NZO/HILtJ, C57BL/6) showed a higher level of cytokine activity (both pro- and anti-inflammatory; *Figure 6a,b and c*) while poorly resilient strains showed either a relatively stunted activity (CAST/EiJ and PWK/PhJ), or tendency towards pro-inflammatory biased expression (NOD/ShiLtJ and A/J). These findings provided explanations at the molecular level that dovetail with our model inference about the variation in the strength and precision of the net effect of immune responses in these strains. Specifically, the model predicted that the highly resilient mouse strains 129S1/SvImJ, NZO/HILtJ and to a large extent C57BL/6, activate the targeted response more strongly than the less resilient CAST/EiJ, NOD/ShiLtJ, and PWK/PhJ. This prediction was consistent with the higher expression of pro-inflammatory cytokines in these strains (*Figure 6a,b and c*), in comparison to two of the lesser resilient strains (CAST/EiJ and PWK/PhJ) for which we predicted weak immune responses (*Figures 4* and *5*). Importantly, in the three strains that show higher survival (129S1/SvImJ, NZO/HILtJ, and C57BL/6), the robust expression of TNF-α and IFN-γ was matched by equally robust expression of IL-10 and TGF-β (*Figure 6d*), which inhibit overproduction of immune effectors (*Artavanis-Tsakonas et al., 2003*). Less resilient CAST/EiJ and PWK/PhJ showed a comparatively low expression of both pro- and anti-inflammatory cytokines (*Figure 6a,b and c*). The lower pro-inflammatory cytokine expression draws a parallel with our modelling results that CAST/EiJ and PWK/PhJ show comparatively weaker immune activation (*Figure 4*).

We found signs of pro-inflammatory bias in the cytokine expressions of NOD/ShiLtJ and A/J (*Figure 6*), both of which are poorly resilient at 40% and 20% survival, respectively (*Figure 1*). In these strains, we observed elevated expression of pro-inflammatory cytokines (particularly IFN-γ) without a matched increase in anti-inflammatory responses (*Figure 6b,c and d*), which inhibit overproduction of immune effectors (*Artavanis-Tsakonas et al., 2003*). For the A/J mice, our model predicted strong immune responses, both indiscriminate and targeted (high $\psi_{N_1}$ and $\psi_{N_2}$; *Figure 4*), consistent with the expectation that a pro-inflammatory bias leads to strong host responses (*King and Lamb, 2015*). However, immunoregulatory imbalance is also associated with immunopathology during malaria infections (*King and Lamb, 2015*). In particular, overproduction of immune effectors could cause collateral host tissue damage and excessive indiscriminate RBC clearance may lead to severe anaemia. Although generally less pronounced than A/J, resilient strains NZO/HILtJ and C57BL/6 also showed a relatively high ratio of pro- to anto-inflammatory cytokines (*Figure 6d*). Our model predicted that AJ, NZO/HILtJ, and C57BL/6 also exhibit strong activation of indiscriminate RBC clearance (high $\psi_{N_1}$; *Figure 4*). Thus, our modelling results and cytokine assays together suggest a causal link between the strength of host-driven destruction of RBCs — which may be both beneficial and detrimental to the host (*Wale et al., 2019*; *Perkins et al., 2011*) — and the underlying pro-inflammatory bias (*Figure 6d*). At first glance, our model prediction that NOD/ShiLtJ mice trigger weaker than average immune activation (low $\psi_{N_1}$ and $\psi_{N_2}$; *Figure 4*) appears incongruent with the strong relative expression of pro-inflammatory cytokines (*Figure 6d*). However, NOD mice are documented for immunodeficiencies downstream of inflammatory cytokines: for example, severely reduced natural killer cell activity (*Kataoka et al., 1983*), hyporesponsiveness of macrophages to growth factors and IFN-γ (*Serreze et al., 1993*), and defective development of antigen-presenting dendritic cells (*Pearson et al., 2003*). Thus, our model prediction and cytokine assay together indicate that

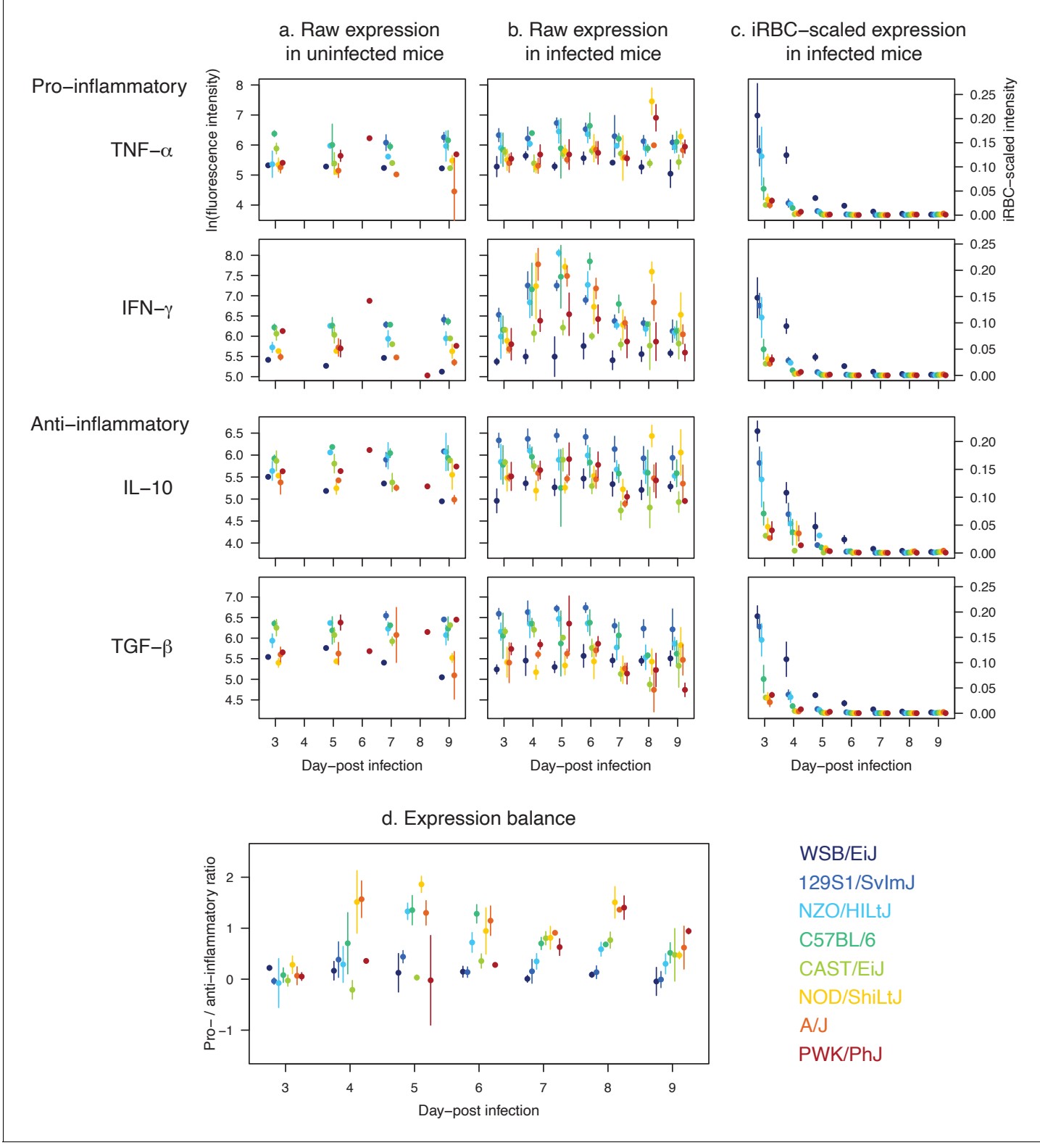

**Figure 6.** Distinct expression patterns of pro (TNF-α and IFN-γ) and anti-inflammatory cytokines (IL-10 and TGF-β) in eight mouse strains infected with *P. chabaudi* show that host resilience to malaria infection is linked to the strength and balance in cytokine expressions. Raw temporal expression intensity in (a) uninfected control and (b) infected mice between day 3 and 9 post-infection. (c) The intensity of cytokine expression scaled by the median iRBC density of the strain per day. A higher value indicates higher propensity to express cytokines against the same density of parasites on a given day. (d) The ratio of pro- to anti-inflammatory cytokine expressions. Shown are additive expressions (i.e. TNF-α + IFN-γ and IL-10 + TGF-β), but

*Figure 6 continued on next page*

*Figure 6 continued*

multiplicative expression patterns (i.e. TNF-α × IFN-γ and IL-10 × TGF-β) were qualitatively identical (results not shown). The points and error bars are the means and standard deviations, respectively. For each day, the strains are ordered from left to right according to host survival as listed in *Figure 1*.

inflammatory responses at the molecular level failed to translate functionally to effective clearance of malaria parasites.

We observed low raw intensity of both pro- and anti-inflammatory cytokines in WSB/EiJ (*Figure 6b*), for which our model predicted exceptional parasite control (through small parasite burst size and high parasite clearance, low β and high $\psi_{N_2}$) and resource suppression (heightened RBC mortality, high $\mu'_R$; *Figures 4* and *5*). At first, low cytokine intensity appeared to be at odds with high parasite clearance. However, it is worth highlighting that the WSB/EiJ mice experience exceptionally low iRBC density (*Figure 2*) and hence the cue for triggering cytokine production remains low. When scaled by the iRBC density, it becomes apparent that the propensity of WSB/EiJ to express both pro- and anti-inflammatory cytokines generally exceeds that of other resilient strains (*Figure 6d*). Further studies are needed to empirically determine what explains the exceptionally low iRBC density in this strain. However, our modelling results suggest that mechanisms other than inflammation may also facilitate WSB/EiJ's resilience: for example, small parasite burst size, low β, may be linked to natural RBC resistance to parasite invasion (*Taylor and Fairhurst, 2014*).

Overall, cytokine assays complemented mathematical modelling by uncovering variation in molecular mechanisms that underlies functional differences among host strains that show diverse infection outcomes. Host resilience to *Plasmodium* infection was linked to a balanced expression of pro- and anti-inflammatory cytokines. Poorly resilient strains either showed stunted activation of pro-inflammatory cytokines associated with insufficient parasite control, or pro-inflammatory bias that has been implicated in immunopathology (*Artavanis-Tsakonas et al., 2003*).

## Conclusion

It is well documented that immune responses are a key host factor influencing protection from malaria infections (*López et al., 2010*). However, it remains difficult to relate health outcomes to the underlying variation in host immunity because parasite load, immune regulation and host health are intertwined and are variable over time. Over the past two decades, several dynamical models have inferred mechanisms of complex within-host ecological interactions from longitudinal data of parasite load and RBCs (i.e. parasite resource and indicator of host health) in the rodent malaria system (e.g. *Haydon et al., 2003*; *Mideo et al., 2008b*; *Kochin et al., 2010*; *Miller et al., 2010*; *Mideo et al., 2011*; *Metcalf et al., 2011*; *Metcalf et al., 2012*; *Santhanam et al., 2014*; *Wale et al., 2019*; *Kamiya et al., 2020*). However, a knowledge gap exists between what is empirically measurable (at the molecular and cellular level, for example immune molecules and cells) to what is functionally important to infection dynamics (at the within-host ecological level, for example net effect of parasite clearance). At the within-host ecological level, our dynamical modelling of infection revealed that better host survival during malaria infection was associated with precisely targeted, robust clearance of blood-stage parasites. Using cross-sectional cytokine assays, we uncovered well-regulated inflammatory cytokine expressions are key molecular signatures of inbred mouse strains that survive malaria infections. By augmenting mathematical modelling of within-host ecology with cross-sectional cytokine assays, our study narrows the gap between functional and molecular mechanisms of host resilience to malaria infection.

## Materials and methods

### Data
#### Mouse strains
Mice were purchased from Jackson Laboratories (WSB/EiJ stock #001145), 129S1/SvImJ stock #002448, NZO/HILtJ stock #002105, CAST/EiJ stock #000928, A/J stock #000646, NOD/ ShiLtJ stock #001976, and PWK/PhJ stock #003715 and Charles River (C57BL/6). A subset of mice (WSB/ EiJ, NZO/HILtJ, and PWK/PhJ) were also bred in-house at Stanford University. Animals were housed

in the Stanford Research Animal Facility according to Stanford University guidelines. All mouse experiments were approved by the Stanford Administrative Panel on Laboratory Care (APLAC).

## Infection with *P. chabaudi*

We administered the AJ strain of *Plasmodium chabaudi* to the experimental animals at a dose of $10^5$ iRBCs and monitored infections longitudinally, for 15 days, as previously described (*Torres et al., 2016*). RBCs were quantified using a BD Accuri C6 Plus cytometer. We quantified parasitemia (i.e. proportion of RBCs infected) via thin blood smears and manual microscope counting. We report parasite density as the number of iRBCs per microliter of blood, which was calculated by multiplying parasitemia by the number of total RBCs per microliter of blood. Survival was monitored daily until day 15 post-infection.

## Cross-sectional cytokine assay

We infected mice with *P. chabaudi* as described above. For cross-sectional sampling, between three and five infected mice of each strain were euthanised each day between 10 am and 2 pm from days 3 to 12 post-infection. For each mouse strain, two uninfected control animals were euthanised at baseline and generally on odd-numbered days. Following euthanasia, 75 microliters of plasma was used for immunoassay using the mouse 38-plex kit (eBiosciences/Affymetrix). Further details are available from *Davis et al., 2021*.

## **Model**

### Dynamical model of malaria asexual cycle

In the experiment, mice were inoculated with iRBCs. Synchronously at midnight, the initial cohort of iRBCs rupture, releasing a new generation of merozoites into the bloodstream (*Paul et al., 2003*). These merozoites then rapidly invade further RBCs where the parasites develop to release the next generation of merozoites with a 24 hr interval. We assumed that regulation of immune responses takes place continuously throughout the day ($0 < t \leq 1$ where $t$ is the fraction of a day) and demographic processes of the host and parasite (i.e., turnover of RBCs, iRBC bursting and RBC invasion by merozoites) happen at the end of each day at midnight ($t = 1$).

### Regulation of host responses

Malaria infection triggers a variety of host responses (*Stevenson and Riley, 2004*; *Price et al., 2001*; *Castro-Gomes et al., 2014*), of which we considered two previously identified as the most quantitatively important: indiscriminate clearance of RBCs and targeted clearance of iRBCs (*Miller et al., 2010*). Similar to previous studies (*Kochin et al., 2010*; *Kamiya et al., 2020*), consider a set of ordinary differential equations tracking the change in the rate of activity of each response $N_i$, where $i$ indicates the type of response (i.e. general RBC clearance, $i = 1$; targeted iRBC clearance, $i = 2$):

$$\frac{dN_{i(t)}}{dt} = \psi_{N_i} \frac{I_{(t)}}{I_{max}} - \phi_{N_i} N_{i(t)}, \tag{1}$$

where $\psi_{N_i}$ and $\phi_{N_i}$ are the activation and decay strength of $N_i$, respectively. Assuming that the abundance of iRBCs reflects that of pathogen-associated molecular patterns (PAMPs), we defined the scaled density of iRBCs, $\frac{I_{(t)}}{I_{max}}$, as the within-host cue driving these responses, where $I_{max}$ is the maximum observed iRBC density in the experiment. We assumed that there is no activity in the absence of infection, consequently there is no constitutive immunity in this model. Because demographic events are formulated in discrete time with a unit of one day, the iRBC density on day $d$, is assumed a constant during the time scale of immune regulation (i.e. $0 < t \leq 1$). Previous estimates indicate that the response activity of indiscriminate clearance decays in approximately one day while the targeted response decays with a half-life an order of magnitude longer than the duration of the acute phase of infection (*Kamiya et al., 2020*): mean half-life of 0.96 and 332.6 days, respectively. Based on these estimates, we made the following simplifying assumptions, eliminating parameters $\phi_{N_1}$ and $\phi_{N_2}$: the indiscriminate activity decays at one day following the Dirac-delta distribution and the targeted activity does not decay during the acute phase. Consequently, we can reformulate *Equation 1*

in discrete-time, assuming that the former response resets daily while the latter accumulates over multiple days without any decay:

$$N_{1(d,t=1)} = \psi_{N_1} \times \frac{I_{(d,t=0)}}{I_{max}} \tag{2}$$

$$N_{2(d,t=1)} = \psi_{N_2} \times \frac{I_{(d,t=0)}}{I_{max}} + N_{2(d,t=0)}. \tag{3}$$

## Turnover of RBCs

The first event at the end of the day ($t = 1$), is clearance and replenishment of RBCs. In the absence of infection, we assumed that RBCs are subjected to background RBC mortality, $\mu_R$. The host replaces RBCs lost to baseline cell mortality by producing $R_c(1 - e^{-\mu_R})$ new RBCs, where $R_c$ is the RBC density at homeostatic equilibrium, assumed equal to the RBC density measured per mouse before parasite inoculation. While we have prior information on baseline mortality and replenishment in the absence of infection (i.e. $\mu_R$; *Van Putten and Croon, 1958*; *Foster et al., 2014*), we fitted two new parameters ($\mu'_R$ and $\mu''_R$, respectively) that allow these processes to be qualitatively different during infection. Our motivation for this was the following. First, during malaria infection experiments, RBCs are lost to daily blood sampling and perhaps handling related stress. Second, fitting $\mu'_R$ allows for the possibility that some indiscriminate clearance of RBCs is independent of iRBC dynamics (unlike in *Equation 2*). Third, there is evidence that erythropoiesis is downregulated during malaria infection (*Wale et al., 2019*), and fitting $\mu''_R$ allowed us to capture this possibility.

Indiscriminate RBC clearance, due to the action of the immune response, occurs at a daily rate $N_{1(d,t=1)}$. In addition to baseline replenishment of RBCs (governed by $\mu''_R$ as described above), RBCs are produced in a density-dependent manner during infection to restore the RBC population (*Chang et al., 2004*) with a time-lag of 2–3 days before the newly produced RBCs are released in the bloodstream (*Savill et al., 2009*). Here assuming a 2-day lag (indicated by $d - 2$), the host produces a fraction ρ of the deviation from RBC density at $R_c$. Infected cells incur an additional rate of mortality, $N_{2(d,t=1)}$ through targeted killing. Together, the turnover of RBCs is expressed as:

$$R_{(d,t=1)} = R_{(d,t=0)} e^{-(\mu'_R + N_{1(d,t=1)})} + R_c(1 - e^{-\mu''_R}) + \rho(R_c - (R_{(d-2,t=1)} + I_{(d-2,t=1)})) \tag{4}$$

$$I_{(d,t=1)} = I_{(d,t=0)} e^{-(\mu'_R + N_{1(t=1)} + N_{2(d,t=1)})}, \tag{5}$$

where $R_{(d,t=1)}$ and $I_{(d,t=1)}$ are the post-turnover densities.

## iRBC bursting

Given synchronous iRBC bursting and the short lifespan of merozoites relative to the length of a day, we modelled iRBC bursting and merozoite invasion as instantaneous events. As iRBCs rupture and release merozoites into the bloodstream at midnight ($t = 1$), the density of merozoites, $M_{(d,t=1)}$ equals $\beta I_{(d,t=1)}$ where β is the parasite burst size per iRBC.

## RBC invasion by merozoites

Upon release, a merozoite either invades an uninfected red blood cell (uRBC), $R_{(d,t=1)}$, at a per capita invasion rate $p$, or it gets cleared before invasion, with a short halflife of $1/\mu_M$ (~30 min [*McAlister, 1977*]). For simplicity, we ignore infections of RBCs by multiple merozoites. Thus, the probability that a given merozoite successfully invades an uRBC is:

$$\frac{p R_{(d,t=1)}}{p R_{(d,t=1)} + \mu_M}. \tag{6}$$

Multiplying the probability by the density of merozoites, and dividing by $R_{(d,t=1)}$, the average number of invading merozoites per uRBC, λ is:

$$\lambda = \frac{M_{(d,t=1)}}{R_{(d,t=1)} + \frac{\mu_M}{p}}. \tag{7}$$

We assumed that the probability of RBC invasion by merozoites is Poisson-distributed with parameter $\lambda$, that is, $\mathrm{Prob}(\text{invasion by } k \text{ mezoroites}) = \frac{\lambda^k e^{-\lambda}}{k!}$ (*Miller et al., 2010*; *Mideo et al., 2011*). Thus, the probability that a given uRBC gets invaded by a merozoite (i.e. $k = 1$) is $\lambda\,e^{-\lambda}$ and the probability that an uRBC escapes merozoite invasion altogether (i.e. $k = 0$) is $e^{-\lambda}$. Ignoring infections of RBCs by multiple merozoites, it follows that the numbers of uRBCs and iRBCs after merozoite invasion (i.e. $R^*_{(d,t=1)}$ and $I^*_{(d,t=1)}$ with the asterisk denoting the post-invasion densities) are:

$$R^*_{(d,t=1)} = R_{(d,t=1)}\,e^{-\lambda} \tag{8}$$

$$I^*_{(d,t=1)} = R_{(d,t=1)}\lambda\,e^{-\lambda}. \tag{9}$$

## Hierarchical Bayesian inference

We fitted the above within-host infection model to the corresponding longitudinal data from 80 mice using a Bayesian statistical approach, which allows for parameter estimation in high dimensional spaces, for example, in hierarchical models where observations are organised in multiple levels of sampling units (*Mugglin et al., 2002*; *Cressie et al., 2009*). In this study, there are two levels of sampling units: mouse strains and subjects (i.e. individual mice).

### Strain-specific and individual variation

We estimated host strain- and individual-specific effects in a set of nine fitted parameters describing within-host ecological processes: that is, samples $s$ and $u$ from $\mathcal{N}(0, \sigma_s)$ and $\mathcal{N}(0, \sigma_u)$, respectively. Below, we collectively refer to the parameter set as $\theta$ ($\theta \ni \mu'_R, \mu''_R, \rho, \psi_{N_1}, \psi_{N_2}, \beta$). The prior distributions for these parameters are provided in *Table 1*.

### Likelihood

A Bayesian approach requires a likelihood function to assess the probability of observing the data given model parameters and associated predictions. Our log-likelihood function assumed that the measurement error for the total density of RBCs (i.e. sum of uRBCs and iRBCs), and iRBCs is distributed normally and $log_{10}$-normally, respectively (*Mideo et al., 2008b*; *Mideo et al., 2011*):

$$\ln L = \sum_{i}^{n_{\mathrm{mice}}} \Big\{ \sum_{d}^{n_{\mathrm{days}}} \ln\Big\{ \frac{1}{\sigma_{\mathrm{RBC}}\sqrt{2\pi}} exp\Big[ -\frac{(D_{i,d}^{\mathrm{RBC}} - M_{i,d}^{\mathrm{RBC}})^2}{2(\sigma_{\mathrm{RBC}})^2} \Big] \Big\} \\ + \sum_{d}^{n_{\mathrm{days}}} \ln\Big\{ \frac{1}{\sigma_{\mathrm{iRBC}}\sqrt{2\pi}} exp\Big[ -\frac{(\log_{10}(D_{i,d}^{\mathrm{iRBC}} + 1) - \log_{10}(M_{i,d}^{\mathrm{iRBC}} + 1))^2}{2(\sigma_{\mathrm{iRBC}})^2} \Big] \Big\} \Big\} \tag{10}$$

where $D_{i,d}^{\mathrm{RBC}}$ and $D_{i,d}^{\mathrm{iRBC}}$ are the observed count of total RBCs and iRBCs, $M_{i,d}^{\mathrm{RBC}}$ and $M_{i,d}^{\mathrm{iRBC}}$ are the model predictions of total RBCs and iRBCs for individual $i$ at day $d$. We estimated standard deviations, $\sigma_{\mathrm{RBC}}$ and $\sigma_{\mathrm{iRBC}}$ for the total RBC and iRBC count, respectively, with specific informative priors (*Mideo et al., 2008b*; *Miller et al., 2010*; *Table 1*). Our modelling focused on the first wave of infection, thus we fitted data up to two weeks post-infection ($n_{\mathrm{days}} = 14$ at maximum). In mice that succumbed to infection, we fitted the model to data until the last sampling prior to death.

### MCMC sampling

Estimating the posterior probability density of parameters of a complex model requires a Markov Chain Monte Carlo (MCMC) sampling algorithm. Our model was written in Stan 2.21.2 and fitted through the RStan interface (*Carpenter et al., 2017*; *Stan Development Team, 2019*), which provides an efficient, general-purpose MCMC sampler (No-U-Turn Hamiltonian Monte Carlo) and a Bayesian inference environment. The model was fitted in parallel in four independent chains, each with 3000 sampled iterations and 1000 warmup iterations. For diagnostics, we confirmed over 400 effective samples and ensured convergence of independent chains using the $\hat{R}$ metric (values below 1.1 are considered an indication of multi-chain convergence) for all parameters (*Gelman et al., 2013*; *Stan Development Team, 2018*). We assessed the goodness of fit to data using standardised residuals (Appendix 1). We also quantified the posterior z-score and posterior contraction to examine the accuracy and precision of posterior distributions, and the relative strength of data to prior information (*Schad et al., 2021*) (Appendix 2).

## Acknowledgements

We thank members of the Mideo group at University of Toronto and the Schneider group at Stanford University for discussions. This work was funded by Mitacs (Globalink Research Exchange Award), IDEAS (RCN Exchange Award), NSERC (Discovery Grant), Compute Canada (Resource Allocation Competition Award), T32 AI007328 and NSF GRFP (NMD), DARPA W911NF-16–0052 (DSS). We are also grateful to the senior editor, Prof. Miles Davenport and three anonymous reviewers for their helpful comments.

## Additional information

### Funding

| Funder | Grant reference number | Author |
| --- | --- | --- |
| Mitacs | Globalink | Tsukushi Kamiya |
| National Science Foundation | IDEAS RCN exchange grant | Tsukushi Kamiya |
| Natural Sciences and Engineering Research Council of Canada | Discovery Grant | Nicole Mideo |
| Compute Canada | Resource Allocation Competition Award | Tsukushi Kamiya Nicole Mideo |
| National Institutes of Health | T32 T32 AI00 | Nicole M Davis David Schneider |
| National Science Foundation | GRFP | Nicole M Davis |
| Defense Advanced Research Projects Agency | W911NF-16-0052 | David Schneider |

The funders had no role in study design, data collection and interpretation, or the decision to submit the work for publication.

### Author contributions

Tsukushi Kamiya, Conceptualization, Formal analysis, Funding acquisition, Validation, Investigation, Visualization, Methodology, Writing - original draft, Writing - review and editing; Nicole M Davis, Conceptualization, Resources, Data curation, Validation, Investigation, Methodology, Writing - original draft, Writing - review and editing; Megan A Greischar, Conceptualization, Supervision, Methodology, Writing - review and editing; David Schneider, Conceptualization, Resources, Supervision, Funding acquisition, Writing - review and editing; Nicole Mideo, Conceptualization, Resources, Supervision, Funding acquisition, Project administration, Writing - review and editing

### Author ORCIDs

Tsukushi Kamiya (iD) https://orcid.org/0000-0002-9020-6699
Megan A Greischar (iD) http://orcid.org/0000-0002-7521-9344
David Schneider (iD) http://orcid.org/0000-0002-2391-9963

### Ethics

Animal experimentation: Animals were maintained specific-pathogen free (SPF) and housed in the Stanford Research Animal Facility according to Stanford University guidelines, accredited by the Association of Assessment and Accreditation of Laboratory Animal Care (AAALAC) International. All mouse experiments were approved by the Stanford Administrative Panel on Laboratory Care (APLAC) under protocol #30923, and every effort was made to minimize animal suffering.

### Decision letter and Author response

Decision letter https://doi.org/10.7554/eLife.65846.sa1
Author response https://doi.org/10.7554/eLife.65846.sa2

## Additional files

### Supplementary files
• Transparent reporting form

### Data availability

We include the entire workflow (data and modelling, analysis and visualisation programmes) as Supplementary file 1. Further details of the data can be found in Davis et al., 2021 (*mBio*; https://doi.org/10.1128/mBio.02424-21).

The following datasets were generated:

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

## Appendix 1

### Assessment of model fit

Our fitted model accurately describes the daily time course of RBCs and iRBCs during the acute phase of malaria infection in mice (*Appendix 1—figure 1*).

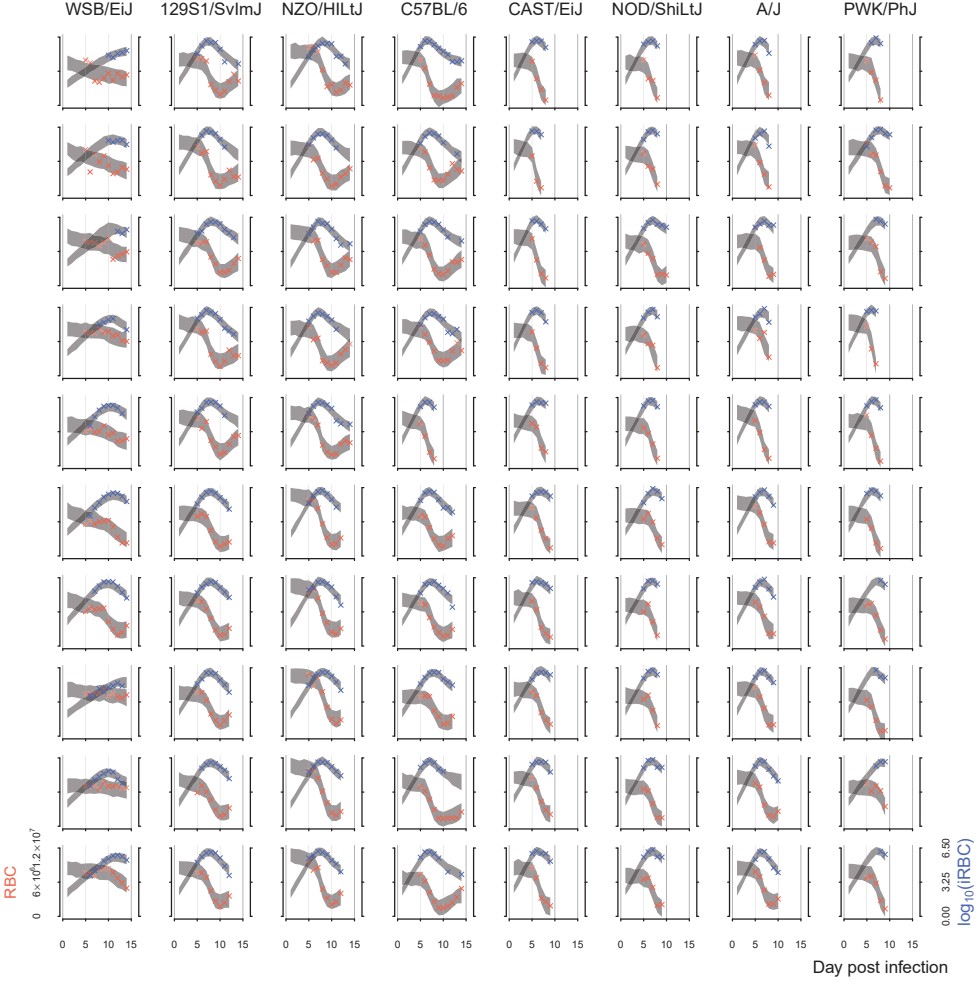

**Appendix 1—figure 1.** The fit of the dynamical model to the density of RBCs (red) and iRBCs (blue). Each column corresponds to a mouse strain. The crosses indicate data and grey bands correspond to 95% predictive intervals of the model, incorporating uncertainty in parameter estimation and sampling.

To provide a rigorous assessment of the model fit, we examined the standardised residuals for RBC and iRBC densities following *Miller et al., 2010*. By integrating over the probability density of each parameter, $\Phi$, the marginal standardised residual of each data point $i$ was defined as:

$$r_{x,i} = \frac{1}{\sigma_x} \int_{\Phi} (x_{data,i} - x_{model,i}(\Phi))d\Phi \tag{11}$$

where $\sigma_x$ is standard deviation of $x$, which is either RBC or iRBC density. The fit of the dynamical model to RBC and iRBC density was accurate without a significant sign of bias (*Appendix 1—figure 2*).

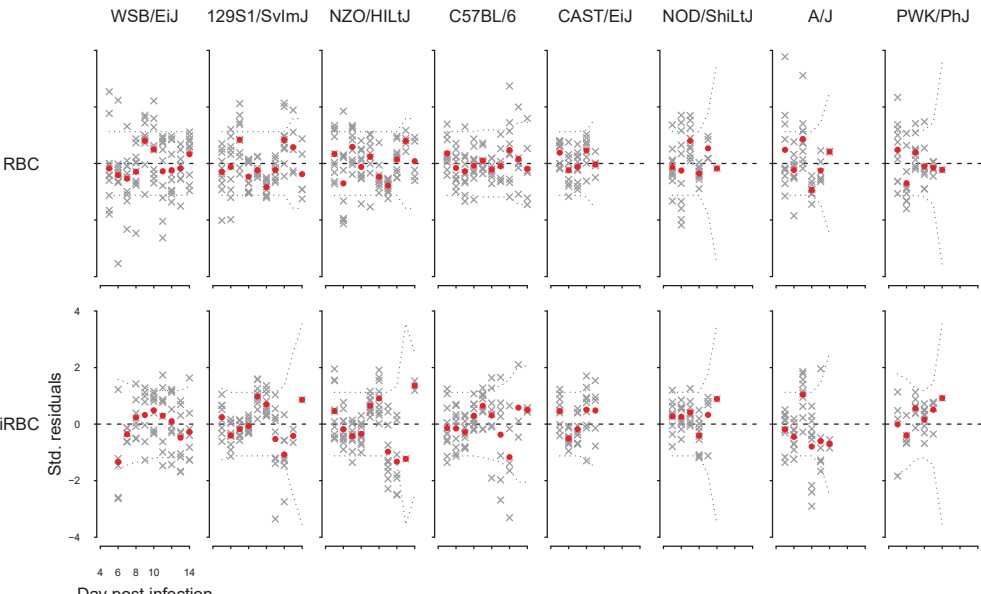

**Appendix 1—figure 2.** Standardised model residuals of the dynamical model. Crosses indicate residuals for individual time series, while red dots indicate the mean; blue dotted lines indicate the Bonferroni-corrected 95% confidence intervals. Poor fits are indicated by the mean residuals deviating from confidence intervals.

## Appendix 2

### Assessment of posterior and prior distributions

Posterior correlations

Visualising correlations between model parameters aids assessing practical identifiability of a parameter set (*Gabry and Mahr, 2021*), i.e., whether the likelihood function can distinguish between the contributions from multiple parameters in the set. The absolute correlation coefficients of 0.7 or larger are generally considered to indicate severe collinearity that adversely affects parameter estimation (*Dormann et al., 2013*). With no such strong correlation observed, the pairwise correlation plot indicates that the MCMC algorithm can sample the parameter set θ without heavy reliance on prior information (*Appendix 2—figure 1*).

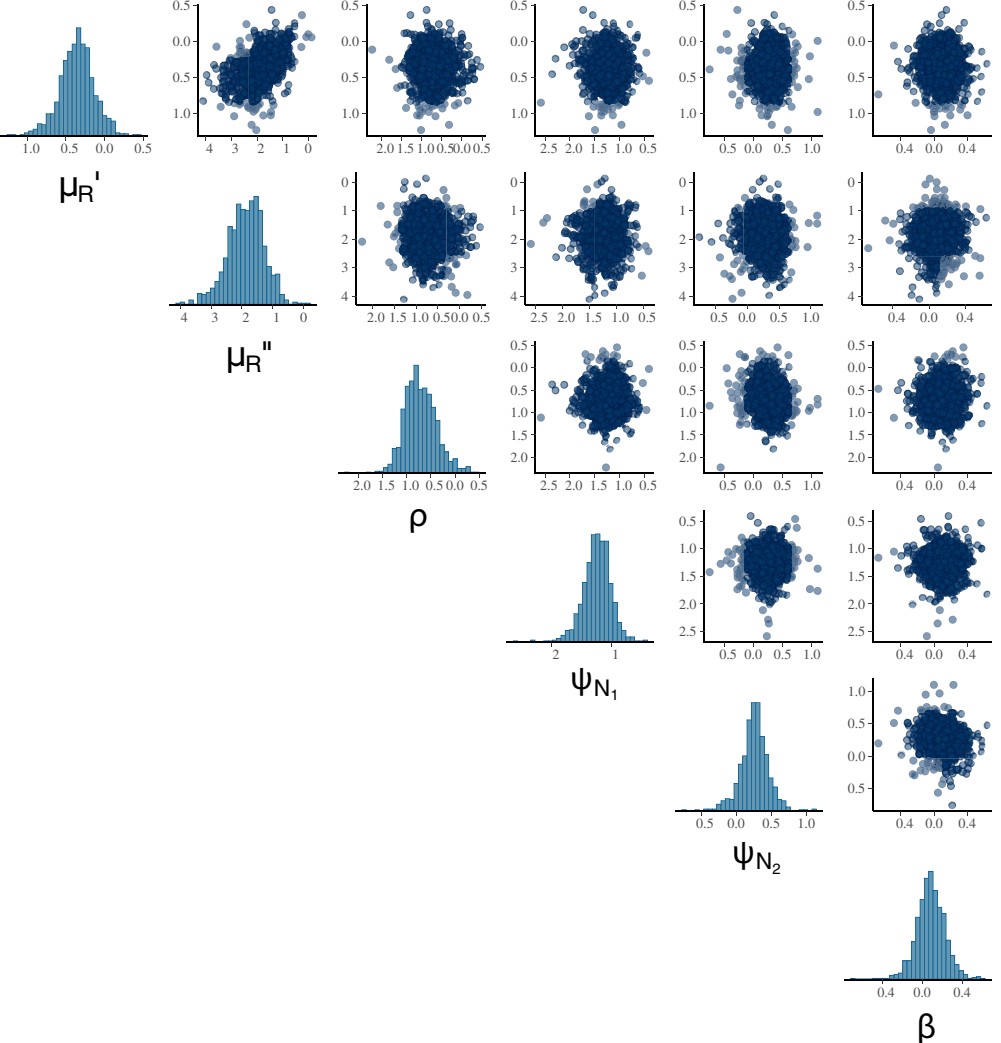

**Appendix 2—figure 1.** Pairwise correlations of MCMC samples indicate that the parameter set $\theta \ni (\mu'_R, \mu''_R, \rho, \psi_{N_1}, \psi_{N_2}, \beta)$ is likely identifiable. The highest correlation coefficient observed was 0.54 between $\mu'_R$ (daily background RBC mortality rate during infection) and $\mu''_R$ (density-independent RBC replenishment rate during infection).

### Posterior accuracy, precision and prior contraction

To provide a rigorous assessment of our Bayesian inference, we leveraged the properties of posterior distributions to interrogate our modelling assumptions. To examine the accuracy and precision

of posterior distributions, we first generated simulated observations based on the estimated posterior mean parameters. We then refitted our model to the simulated observations (i.e., secondary fitting) to compute the posterior z-score for each parameter, which measures how closely the posterior recovers the parameters of the true data generating process (*Schad et al., 2021*):

$$z = \frac{\mathbb{E}_{\text{sim}} - \mathbb{E}_{\text{post}}}{\sigma_{\text{sim}}},$$

where $\mathbb{E}_{\text{post}}$ denotes the posterior mean of the fit to the actual data that we consider the 'true' parameter. $\mathbb{E}_{\text{sim}}$ and $\sigma_{\text{sim}}$ denote the mean and standard deviation of the posterior distribution of the secondary fitting. The smaller the absolute z-score, the closer the bulk of the posterior is to the true parameter: z-scores beyond the absolute value of three to four may indicate substantial bias (*Schad et al., 2021*).

To examine the influence of the likelihood function in relation to prior information, we computed the posterior contraction:

$$1 - \frac{\sigma_{\text{post}}^2}{\sigma_{\text{prior}}^2}$$

where $\sigma_{\text{post}}^2$ and $\sigma_{\text{prior}}^2$ correspond to the variance of posterior and prior distributions, respectively. The posterior contraction values close to zero indicate that data contain little information (i.e., poor identifiability, rendering priors strongly informative). Conversely, values close to one indicate that data are much more informative than the prior (*Schad et al., 2021*).

We found that most of our model parameters — i.e., $\theta \ni (\mu_R', \mu_R'', \rho, \psi_{N_1}, \psi_{N_2}, \beta)$ and hyperpriors $\sigma_s$ and $\sigma_u$ — were estimated with accuracy, precision and identifiability, with the absolute posterior z-scores well below three and posterior contraction values beyond 75% for most parameters (*Appendix 2—figure 2*). The strain-level variation, $\sigma_s$ for $\rho$ (proportion of anaemia restored per day) tended towards overfitting with the posterior z-score of $-2.99$ (*Appendix 2—figure 2*). Thus, caution might be warranted when interpreting strain-specific differences in this parameter. One parameter, $\mu_R''$ (density-independent RBC replenishment rate during infection) showed a comparatively lower posterior contraction value, yet its posterior distribution contracted by 64.0%, meaning that data still provided substantial information over the prior distribution (*Appendix 2—figure 2*).

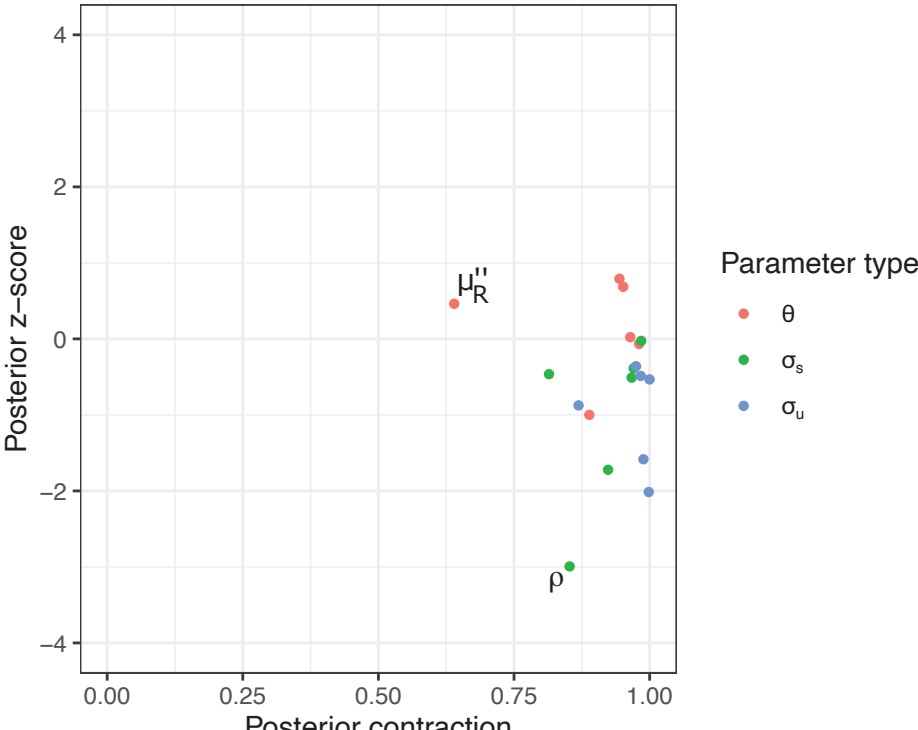

**Appendix 2—figure 2.** The model parameters were estimated with accuracy, precision and identifiability. Posterior z-score (y-axis) measures how closely the posterior recovers the parameters of the true data generating process and posterior contraction (x-axis) evaluates the influence of the likelihood function over the prior, respectively. Smaller absolute posterior z-scores indicate that the posterior accurately recovers the parameters of the true data generating process: the absolute value beyond three to four may indicate substantial bias (*Schad et al., 2021*). The posterior contraction values close to one indicate that data are much more informative than the prior.

