## [Decision Letter]

**Acceptance summary:**

This paper will be of interest to parasitologists, immunologists and modellers working in the field of malaria. It provides a model-based approach to quantify different host responses to malaria infection and identify the response(s) best explaining the infection outcomes (i.e., survival rate and host resilience to malaria infection). The methods for model construction and fitting and data analysis are appropriate and rigorous, however, extra work is required for validation of model predictions, which are critical for the interpretation of the results.

**Decision letter after peer review:**

Thank you for submitting your article "Within-host dynamics reveal functional diversity of host resilience to malaria infection" for consideration by *eLife*. Your article has been reviewed by 3 peer reviewers, including Jennifer Flegg as the Reviewing Editor and Reviewer #1, and the evaluation has been overseen by Miles Davenport as the Senior Editor.

Essential revisions:

1) Parameter correlations. The results of the manuscript are primarily based on the parameter estimates, which are calculated by the samples of the posterior distributions, so the correlations between model parameters (based on the posterior samples) should be examined. One example for the correlated parameter would be the activation rate and decay rate of immune response in equation 1. Since the immune response N_i_ was not measured and used in model fitting, the two parameters should show a strong positive correlation, which would prevent accurate estimates. A clear view of the parameter correlations is critical for the reliability of the estimates and in turn any conclusion based on the estimates.

2) Prior distributions. These were not shown on top of the posteriors such that it is difficult to judge how much information was from the priors and how much was from the data. Since the priors are relatively informative, e.g. 2.5 and sqrt(2.5) (see Table 1), it is also necessary to examine the impact of the choice of the priors on the posteriors by a sensitivity analysis. The choice of priors is not well-justified. How were the hyperparameters for the host response parameters chosen? Is the model sensitive to these?

3) The authors showed that the too strong and too weak immune activation predicted by the model was associated with high host mortality and concluded that a balanced immune response may be a major determinant of host survival. However, a stronger immune activation in the model refers to a higher activation rate (i.e., a higher value of parameter psi in equation 1). This is not necessarily related to a balance immune response, which is a balanced level of expression of pro-inflammatory and anti- inflammatory cytokines and is mathematically interpreted by that a strong/weak activation should be associated with a strong/weak decay such that the net effect (N_i_ in equation 1) is maintained at a healthy level. Further clarification on this point is necessary.

4) Cytokine dynamics. In the results, the authors selected and discussed some model predictions that are consistent with what the cytokine data suggest. However, it lacks a comprehensive comparison between the data and the model prediction. I am wondering if it is possible to create a figure similar to Figure 3b (only psi's and phi's) using the cytokine data and compare with the model prediction. It will tell clearly how well the model predictions are consistent with the data. Could you connect the cytokine dynamics with the mechanistic model more directly? This seems like a missed opportunity – you have beautiful timecourses here – could you reframe the model to directly include processes reflected by the cytokines?

5) Figures. It would be useful to have a detailed graphical representation of the model in the main text. The temporal information in Figure 5 is hard to process.

*Reviewer #1:*

This paper presents a dynamic mathematical model of malaria parasites and red blood cells in a mouse model. A strength of the paper is the collection of experimental data to fit to the mathematical model, however the model is overly simplified and is therefore unlikely to have high impact in the field.

*Reviewer #2:*

Kamiya, Davis et al., studied the associations between the diversity of host responses to malaria infection and infection outcomes (i.e., survival rate and host resilience to malaria infection). In detail, they collected the longitudinal data from mice that are genetically different and estimated some key biological parameters governing the parasite replication dynamics and immune responses by fitting a mathematical model to the data. By comparing the parameter estimates and a set of cytokine data collected from the same pool of mice, they identified some consistency between model predictions and data and concluded that the longitudinal data contain important information that may facilitate the explanation of the diverse infection outcomes. The methods for model fitting and data analysis are appropriate and rigorous. The conclusion about balanced immune response is partly supported by the data but not clearly associated with model predictions. Some further analyses on fitting results are necessary in order to ensure the reliability of some conclusions.

*Reviewer #3:*

The first section of the analysis connects parameter estimates with outcome. With such a large model this section needs a thorough analysis of correlations between parameters – the PCA goes a little way towards this, but not enough. Identifying parameter subspaces might also make the interpretation of these fitted values simpler.

The second section – the data are rich, but it feels let down slightly by its qualitative nature. Could the authors connect the cytokine dynamics with the mechanistic model more directly? This seems like a missed opportunity – there are beautiful timecourses here – could the model be reframed to directly include processes reflected by the cytokines?

It would be useful to have a detailed graphical representation of the model

in the main text.

The temporal information in Figure 5 is hard to process. I like the idea of the 'phase-space' view but here I think it would be clearer to have the timecourses of pro and anti-inflammatory cytokines stacked on top of each other.

At what time(s) of day were the cytokine measurements made? Were they consistent? Given there is a daily replication/infection cycle, I would guess timing matters.

p9 "We found that reducing the expression of anti-inflammatory …" This phrase is misleading as there was no intervention that reduced expression.

---

## [Author Response]

Essential revisions:1) Parameter correlations. The results of the manuscript are primarily based on the parameter estimates, which are calculated by the samples of the posterior distributions, so the correlations between model parameters (based on the posterior samples) should be examined. One example for the correlated parameter would be the activation rate and decay rate of immune response in equation 1. Since the immune response N_i_ was not measured and used in model fitting, the two parameters should show a strong positive correlation, which would prevent accurate estimates. A clear view of the parameter correlations is critical for the reliability of the estimates and in turn any conclusion based on the estimates.

We fully re-examined our modelling assumptions based on this request. First, we have eliminated the decay parameters *ϕ_N1_* and *ϕ*_N2_, which were correlated with the activation parameters ψ_N1_ and ψ_N2_, as the Editor rightly pointed out. Our previous estimates indicate that the response activity of indiscriminate clearance decays approximately in one day while the targeted response decays with a half-life an order of magnitude longer than the duration of the acute phase infection (Kamiya et al., 2020 PLOS Comp Biol). Based on these estimates, we made the simplifying assumptions that the indiscriminate activity decays with a half-life of one day following the Dirac-δ distribution and the targeted activity does not decay during the acute phase. Consequently, we reformulated immune regulation in discrete-time, assuming that the former response resets daily while the latter accumulates over multiple days without any decay (Figure 3a, Equation 2 and 3). These changes are detailed between lines 247 and 253. We acknowledge that it is potentially problematic to rely on previous modelled inferred parameter estimates where reliance on prior information was not systematically quantified. To demonstrate that our estimates are robust despite incorporating estimates from previous modelling studies (namely, Miller et al., 2010, Mideo et al., 2011, and Kamiya et al., 2020), we conducted a rigorous examination of the posterior distributions (i.e., posterior zscore and posterior contraction) to demonstrate that our inference was not biased and prior information had relatively small influence on the posterior compared to data.

Second, we found that the ratio of merozoite mortality to invasion rate showed a moderately high correlation (>0.65) with burst size and did not show substantial strain-specific variation. Thus we have now fixed the ratio based on a previously published estimate (Miller et al., 2010 PLOS Comp Biol).

We confirm that the new simplified model preserves the goodness of fit of the older model (Appendix 1) while markedly improving the MCMC sampling behaviour. We believe that eliminating ϕ_N1_ and ϕ_N2_ does not impact our inference substantially because we found larger strain-specific variability in the rates of activation (ψ_N1_ and ψ_N2_) than decay of responses (ϕ_N1_ and ϕ_N2_) in the previously submitted version. This indicated to us that immunogenic differences between strains that we can infer using our model are largely due to differences in the speed and strength with which responses are turned on, rather than how long those responses last.

Based on this new model, we now provide a pairwise correlation plot, which shows that there are no strong correlations between modelled parameters: the highest correlation (r = 0.54) was found between background RBC mortality rate and density-independent RBC replenishment rate during infection (Appendix 2 Figure 1).

2) Prior distributions. These were not shown on top of the posteriors such that it is difficult to judge how much information was from the priors and how much was from the data. Since the priors are relatively informative, e.g. 2.5 and sqrt(2.5) (see Table 1), it is also necessary to examine the impact of the choice of the priors on the posteriors by a sensitivity analysis. The choice of priors is not well-justified. How were the hyperparameters for the host response parameters chosen? Is the model sensitive to these?

Thanks to eliminating poorly identifiable parameters, our model now samples well with weaker priors in the scale of exp(𝒩(0,1)) compared to exp(𝒩(0,0.25)) in the previously submitted version. We also corrected an oversight in leaving out the hyperpriors from the parameter table (Table 1), which are now estimated with a prior, exp(𝒩(0,1)). This prior is generic and even less informative than recommended by the Stan Developmental Team (https://github.com/stan-dev/stan/wiki/Prior-Choice-Recommendations).

We agree with the Editor that the influence of prior information should be studied and presented. To provide a rigorous assessment of Bayesian inference, we now evaluate properties of posterior and prior distributions by estimating the posterior z-score and posterior contraction (Appendix 2). The former measures how closely the posterior recovers the parameters of the “true” data generating process while the latter quantifies the influence of data in relation to prior information. By plotting the two values against each other (Appendix 2 Figure 2), we can identify problematic posterior behaviours including bad priors, poor identifiability (i.e., strong priors), overfitting (i.e., inability to recover “true” parameters.) (Betancourt 2020 Towards A Principled Bayesian Workflow).

We confirm that we detected no clear sign of problematic posterior behaviours (Appendix 2 Figure 2), though the strain-level variation, σ_s_ for rho (proportion of anaemia restored per day) showed a slight tendency towards overfitting. Our model successfully recovers “true” parameter values from simulated data (i.e., low absolute posterior z-score, – 2.99 at maximum) and posterior distributions are substantially narrower than the prior distributions (high posterior contraction, >75% contraction for all but one and 64% at minimum).

3) The authors showed that the too strong and too weak immune activation predicted by the model was associated with high host mortality and concluded that a balanced immune response may be a major determinant of host survival. However, a stronger immune activation in the model refers to a higher activation rate (i.e., a higher value of parameter psi in equation 1). This is not necessarily related to a balance immune response, which is a balanced level of expression of pro-inflammatory and anti- inflammatory cytokines and is mathematically interpreted by that a strong/weak activation should be associated with a strong/weak decay such that the net effect (N_i_ in equation 1) is maintained at a healthy level. Further clarification on this point is necessary.

In the previously submitted version, we considered our cytokine results as “confirmatory” of our mathematical modelling. Instead, this comment helped us realise that the cytokine results actually offer “complementary” insights at different scales: the cytokine data provide molecular explanations of the differences in the net effect of host responses we predicted using the mathematical model.

We agree with the editor that a higher immune activation rate does not relate directly to balanced immunity. Our model predicted a robust targeted response for resilient strains while a weak targeted response (CAST/EiJ, NOD/ShiLtJ, PWK/PhJ) or a too strong indiscriminate response (A/J) for poorly resilient strains. Thus, we now interpret the primary motif for resilience as the strength and precision of parasite clearance. Cytokine assays then revealed that a strong and precise net effect of immune response was underpinned by strong and balanced expressions of pro- and anti-inflammatory cytokines while poorly resilient strains lacked either pro- (CAST/EiJ, PWK/PhJ) or anti-inflammatory cytokine (A/J). Thus, we now distinguish that our inference about the role of balanced immunity was derived from cytokine assays, and not mathematical modelling. To clarify this nuanced distinction, we changed the phrasing from “balance” to “precision” when discussing our modelling results.

4) Cytokine dynamics. In the results, the authors selected and discussed some model predictions that are consistent with what the cytokine data suggest. However, it lacks a comprehensive comparison between the data and the model prediction. I am wondering if it is possible to create a figure similar to Figure 3b (only psi's and phi's) using the cytokine data and compare with the model prediction. It will tell clearly how well the model predictions are consistent with the data. Could you connect the cytokine dynamics with the mechanistic model more directly? This seems like a missed opportunity – you have beautiful timecourses here – could you reframe the model to directly include processes reflected by the cytokines?

As we mention in the reply above, we realise that our previous assertion that the cytokine data offer direct validation of the model predictions was misleading, and corrected this error. We also agree with the Editor that the temporal aspect of our cytokine data was not communicated well. To improve, we now provide a new figure, Figure 6 to better demonstrate the temporal change in the cytokine expressions. We note that this figure is not meant to offer a direct comparison of ψ_N1_ and ψ_N2_ in Figure 4b as cytokine data offer insights at a different scale (i.e., molecular) from the modelling result (i.e., within-host ecology).

In part, the model parameters like ψ_N1_ and ψ_N2_ are not directly comparable to observable phenotypes because variation in ψ_N1_ and ψ_N2_ represents the intrinsic differences independent of within-host cues (e.g., infected red blood cell density). In contrast, cytokine expression is a product of the intrinsic differences and the changing within-host cues. Within-host cues are also different among host strains. To facilitate the comparison among strains, we now provide a figure of cytokine expressions scaled by the iRBC density per day (Figure 6c). This new figure demonstrates, for example, that the cytokine intensity of the WSB/EiJ strain is comparable with the other three resilient strains once standardised for its low parasite density.

Even after controlling for variation in within-host cues, our modelling of within-host ecology is not designed to correspond directly with cytokine expressions as they are emergent properties of a complex network, of which cytokines take part as regulators (i.e., upstream of effector responses). For example, we observed high pro-inflammatory cytokine expression in NOD/ShiLtJ, despite our prediction of low immune responses. This discrepancy is likely due to previously documented immunodeficiency downstream at the cellular level in this strain. We believe that this difference highlights a unique merit of Bayesian inference using the within-host ecology data in revealing complex traits like the strength of response activation, which directly relates to infection dynamics and outcomes, yet it is difficult to measure directly.

We agree that it would be desirable to quantitatively link cytokine expressions to infection dynamics, and ultimately infection outcomes. We recognise two potential approaches. However, we foresee major limitations given our dataset and the current state of knowledge of immune interactions. First, as the editor suggests, it may seem reasonable to correlate the modelling outputs with cytokine data. A key limitation of the correlation approach is that longitudinal data used to fit the model and cytokine data come from different mice as the latter required mouse sacrifice to collect enough blood. As such one would have to reduce the data to the strain level (a maximum of N=8 data points), which severely limits our power to detect meaningful correlations. To clarify this point, we now mention that cytokine data were sampled destructively earlier in the manuscript (L139-141).

The second, and a more conceptually pleasing approach would be to directly integrate cytokine interactions in a mechanistic modelling framework. A major conceptual challenge here is that we lack a complete map of molecular and cellular interactions that generates the net effect of host responses that impacts within-host ecology. Also, we have access to data on only a small number of known molecular components. As such, a priori, we could only construct a partial, sparsely observed network of molecular and cellular interactions that drives the process of parasite clearance. A case study demonstrates that Bayesian estimation of network interactions is plagued by poor identifiability when network regulatory nodes are unobserved (https://www.martinmodrak.cz/2018/05/14/identifying-non-identifiability/). As such, we believe that neither our dataset nor modelling approach is suitable for modelling a network of molecular interactions directly at this time.

We believe that the combination of modelling and cytokine assays among diverse genetic backgrounds offers unique qualitative insights that narrow the gap between diverse infection dynamics (i.e., within-host ecology) and their molecular underpinnings.

5) Figures. It would be useful to have a detailed graphical representation of the modelin the main text.

We added a graphical schematic of our mathematical model (Figure 3).

The temporal information in Figure 5 is hard to process.

We now provide a longitudinal representation of this data (new Figure 6a).